# Robust Gaussian Covariance Estimation in Nearly-Matrix Multiplication Time

**Jerry Li**
Microsoft Research
jerrl@microsoft.com

**Guanghao Ye**
University of Washington
ghye@uw.edu

## Abstract

Robust covariance estimation is the following, well-studied problem in high dimensional statistics: given $N$ samples from a $d$-dimensional Gaussian $\mathcal{N}(\mathbf{0}, \Sigma)$, but where an $\varepsilon$-fraction of the samples have been arbitrarily corrupted, output $\widehat{\Sigma}$ minimizing the total variation distance between $\mathcal{N}(\mathbf{0}, \Sigma)$ and $\mathcal{N}(\mathbf{0}, \widehat{\Sigma})$. This corresponds to learning $\Sigma$ in a natural affine-invariant variant of the Frobenius norm known as the *Mahalanobis norm*. Previous work of [CDGW19] demonstrated an algorithm that, given $N = \Omega(d^2/\varepsilon^2)$ samples, achieved a near-optimal error of $O(\varepsilon \log 1/\varepsilon)$, and moreover, their algorithm ran in time $\widetilde{O}(T(N, d) \log \kappa / \operatorname{poly}(\varepsilon))$, where $T(N, d)$ is the time it takes to multiply a $d \times N$ matrix by its transpose, and $\kappa$ is the condition number of $\Sigma$. When $\varepsilon$ is relatively small, their polynomial dependence on $1/\varepsilon$ in the runtime is prohibitively large. In this paper, we demonstrate a novel algorithm which achieves the same statistical guarantees, but which runs in time $\widetilde{O}(T(N, d) \log \kappa)$. In particular our runtime has no dependence on $\varepsilon$. When $\Sigma$ is reasonably conditioned, our runtime matches that of the fastest algorithm for covariance estimation without outliers, up to poly-logarithmic factors, showing that we can get robustness essentially "for free."

## 1 Introduction

Covariance estimation is one of the most fundamental high dimensional statistical estimation tasks, see e.g. [BL+08a, BL+08b], and references therein. In this paper, we study the problem of covariance estimation in high dimensions, in the presence of a small fraction of adversarial data. We consider the following standard generative model: we are given samples $X_1, \ldots, X_N$ drawn from a Gaussian $\mathcal{N}(\mathbf{0}, \Sigma)$, but an $\varepsilon$-fraction of these points have been arbitrarily corrupted. The goal is then to output $\widehat{\Sigma}$ minimizing the total variation distance between $\mathcal{N}(\mathbf{0}, \Sigma)$ and $\mathcal{N}(\mathbf{0}, \widehat{\Sigma})$. As we shall see, this naturally corresponds to learning $\Sigma$ in an affine-invariant version of the Frobenius norm, known as the *Mahalanobis norm* (see Section 2).

In the non-robust setting, where there are no corruptions, the problem is well-understood from both a information-theoretic and computational perspective. It is known that the empirical covariance of the data converges to the true covariance at an optimal statistical rate: the empirical covariance matrix has expected Mahalanobis error at most $O(d/\sqrt{N})$; and this is the optimal bound up to a constant factor. That is, when we have $N = \Omega(d^2/\varepsilon^2)$, the empirical covariance matrix will have Mahalanobis error $O(\varepsilon)$. In fact, it satisfies a stronger and more natural affine-invariant error guarantee, as we will discuss later in this Section. Moreover, it is easy to compute: it can be computed in time $T(N, d)$, where $T(n, m)$ is the time it takes to multiply a $m \times n$ matrix by its transpose. When $N = \Theta(d^2/\varepsilon^2)$,

the currently known best runtime for this is $\widetilde{O}(Nd^{1.252})$ [GU18].[1] Moreover, this runtime is unlikely to improve without improving the runtime of rectangular matrix multiplication.

The situation is a bit muddier in the robust setting. If there are an $\varepsilon$-fraction of corrupted samples, the information-theoretically optimal error for covariance estimation of $\mathcal{N}(\mathbf{0}, \Sigma)$ is $O(\varepsilon + d/\sqrt{N})$. In particular, when $N = \Omega(d^2/\varepsilon^2)$, we can achieve error $O(\varepsilon)$ [Rou85, CGR+18]. However, the algorithms which achieve this rate run in time which is exponential in the dimension $d$. In [DKK+16], the authors gave the first polynomial-time algorithm for this problem, which given enough samples, achieves error which is independent of the dimension. Specifically, they achieve an error of $O(\varepsilon \log 1/\varepsilon)$, which matches the information-theoretic limit, up to logarithmic factors, and is likely optimal for efficient algorithms [DKS17], up to constants. However, their sample complexity and runtime—while polynomial—are somewhat large, and limit their applicability to very large, high dimensional datasets. More recently, [CDGW19] gave an algorithm which runs in time $\widetilde{O}(T(N, d)/\varepsilon^8)$. When $\varepsilon$ is constant, the runtime of this algorithm nearly matches that of the non-robust algorithm. However, the dependence on $\varepsilon$ is prohibitive for $\varepsilon$ even moderately small. This raises a natural question: *can we obtain algorithms for robust covariance estimation of a Gaussian whose runtimes (nearly) match rectangular matrix multiplication?*

In this paper, we resolve this question in the affirmative. Informally, we achieve the following guarantee:

**Theorem 1** (informal, see Theorem 2). *Let $D$ be a Gaussian distribution with unknown covariance $\Sigma$, where $\Sigma$ has polynomial condition number. Let $0 < \varepsilon < \varepsilon_0$ for some universal constant $\varepsilon_0$. Given a set of $N = \widetilde{\Omega}(d^2/\varepsilon^2)$ samples from $D$, where an $\varepsilon$-fraction of these samples have been arbitrarily corrupted, there is an algorithm that runs in time $\widetilde{O}(T(N, d))$ and outputs $\widehat{\Sigma} \in \mathbb{R}^{d \times d}$ such that the Malahanobis distance between $\Sigma$ and $\widehat{\Sigma}$ is at most $O(\varepsilon \log 1/\varepsilon)$.*

By combining this with the result of [DHL19], this allows us to robustly learn a polynomially-conditioned Gaussian to total variation distance $O(\varepsilon \log 1/\varepsilon)$ in time $\widetilde{O}(T(N, d))$.

Our algorithm follows the same general framework as the algorithm in [CDGW19]. They reduce the problem of covariance estimation given corrupted Gaussian samples $X_1, \ldots, X_N$, to a robust mean estimation problem given samples $Y_i = X_i \otimes X_i$, where $\otimes$ denotes Kronecker product. Then, their algorithm proceeds in two phases: first, they invoke a robust mean estimation algorithm to achieve a rough estimate of the covariance, then they give a procedure which, given a rough estimate of the covariance, can improve it. They show that both steps can be reduced to solving a packing SDP to high accuracy, and invoke black-box nearly-linear time SDP solvers [AZLO15, AZLO16] to obtain their desired runtime. However, both phases incur $\mathrm{poly}(1/\varepsilon)$ running time, because in both cases, they need to solve the packing SDP to $\mathrm{poly}(\varepsilon)$ accuracy, and the black-box packing SDP solvers require $\mathrm{poly}(1/\varepsilon)$ runtime to do so.

Our main contribution is to demonstrate that both phases of their algorithms can be made faster by using techniques inspired by the quantum entropy scoring algorithm presented in [DHL19]. The first phase can be directly improved by using the robust mean estimation in [DHL19] to replace the robust mean estimation algorithm used in [CDGW19] that achieves error $O(\sqrt{\varepsilon})$. Improving the second phase requires more work. This is because the algorithm in [DHL19] for robust mean estimation below error $O(\sqrt{\varepsilon})$ requires that the uncorrupted samples are isotropic, i.e. their covariance is the identity, and have sub-gaussian tails. However, the $Y_i$ are only approximately isotropic, and moreover, have only sub-exponential tails. Despite this, we demonstrate that we can modify the algorithm and analysis in [DHL19] to handle both of these additional complications.

## 1.1 Related work

The study of robust statistics can be traced back to foundational work of Anscombe, Huber, Tukey and others in the 1960s [Ans60, Tuk60, Hub92, Tuk75]. However, it was only recently that first polynomial time algorithms were demonstrated for a number of basic robust estimation tasks, including robust covariance estimation, with dimension-independent (or nearly dimension-independent) error [DKK+16, LRV16]. Ever since, there has been a flurry of work on learning algorithms in the

presence of adversarial training outliers, and a full survey of this literature is beyond the scope of this paper. See recent theses [Li18, Ste18] and the survey [DK19] for a more thorough account.

In particular, we highlight a recent line of work on very efficient algorithms for robust estimation tasks [CDG19, DHL19, LD19, CDGW19, CMY20] that leverage ideas from matrix multiplicative weights and fast SDP solvers. In particular, [CDG19] gave an algorithm for robust mean estimation of a Gaussian in time $\widetilde{O}(Nd/\varepsilon^6)$; this was improved via quantum entropy scoring to $\widetilde{O}(Nd)$ in [DHL19]. Our main contribution is to show that similar techniques can be used to improve the runtime of [CDGW19] to remove the $\text{poly}(1/\varepsilon)$ dependence.

## 2 Formal Problem Statement and Our Results

Here, we formally define the problem we will consider throughout this paper. Throughout this paper, we let $\| \cdot \|_F$ denote the Frobenius norm of a matrix, and $\| \cdot \|$ denote the spectral norm.

**The $\varepsilon$-corruption model**    We will focus on the following, standard corruption model:

**Definition 2.1** ($\varepsilon$-corruption, See [DKK+16]). *Given $\varepsilon > 0$, and a class of distribution $\mathcal{D}$, the adversary operates as follows: The algorithm specifies some number of samples $N$. The adversary generate $N$ samples $X_1, X_2, \ldots, X_N$ from some (unknown) distribution $D \in \mathcal{D}$. The adversary is allowed to inspect the samples, removes $\varepsilon N$ of them, and replaces them with arbitrary points. The set of $N$ points (in any order) is then given to the algorithm.*

Specifically, we will study the following problem: Given an $\varepsilon$-corrupted set of $N$ samples from an unknown $\mathcal{N}(\mathbf{0}_d, \Sigma)$ over $\mathbb{R}^d$, we want to find an accurate estimate of $\Sigma$. Throughout this paper, we will assume that $\varepsilon < c$ for some constant $c$ sufficiently small. The largest $c$ for which our results hold is known as the *breakdown point* of the estimator, however, for simplicity of exposition, we will not attempt to optimize this constant in this paper.

**Mahalanobis distance**    To make this question formal, we also need to define an appropriate measure of distance. As discussed in previous works, see e.g. [DKK+16], the natural statistical measure of distance for this problem is the total variation distance, which we will denote $d_{\mathsf{TV}}(\cdot, \cdot)$. Thus the question is: given an $\varepsilon$-corrupted set of samples from $\mathcal{N}(\mathbf{0}_d, \Sigma)$, output $\widehat{\Sigma}$ minimizing $d_{\mathsf{TV}}(\mathcal{N}(\mathbf{0}_d, \Sigma), \mathcal{N}(\mathbf{0}_d, \widehat{\Sigma}))$. This turns out to be equivalent to learning to unknown covariance in a preconditioned version of the Frobenius norm, which is also often referred to as the *Mahalanobis norm*:

**Fact 2.2** (folklore). *Let $\Sigma, \Sigma'$ be positive definite. Then there exist universal constants $c, C > 0$ so that:*

$$c \cdot \|\Sigma^{-1/2} \Sigma' \Sigma^{-1/2} - I\|_F \leq d_{\mathsf{TV}}(\mathcal{N}(\mathbf{0}_d, \Sigma), \mathcal{N}(\mathbf{0}_d, \Sigma')) \leq C \cdot \min\left(1, \|\Sigma^{-1/2} \Sigma' \Sigma^{-1/2} - I\|_F\right). \tag{1}$$

Thus, the question becomes: given an $\varepsilon$-corrupted set of samples from $\mathcal{N}(\mathbf{0}_d, \Sigma)$, output $\widehat{\Sigma}$ which is as close as possible to $\Sigma$ in Mahalanobis norm.

### 2.1 Our Main Result

With this, we can now state our main result:

**Theorem 2** (Main Theorem). *Let $D = \mathcal{N}(\mathbf{0}_d, \Sigma)$ be a zero-mean unknown covariance multivariate Gaussian over $\mathbb{R}^d$. Let $\kappa$ be the condition number of $\Sigma$. Let $0 < \varepsilon < c$, where $c$ is a universal constant. Let $S$ be a $\varepsilon$-corrupted set of samples from $D$ of size $N = \widetilde{\Omega}(d^2/\varepsilon^2)$. Algorithm 1 that runs in time $\widetilde{O}(T(N, d) \log \kappa)$ takes $S$ and $\varepsilon$, and outputs a $\widehat{\Sigma}$ so that with probability at least $0.99$, we have $\|\Sigma^{-1/2} \widehat{\Sigma} \Sigma^{-1/2} - I\|_F \leq O(\varepsilon \log(1/\varepsilon))$.*

We make several remarks on this theorem.

First, standard reductions (see e.g. [DKK+16]) also allow us to robustly learn the covariance of a Gaussian with arbitrary mean, by doubling $\varepsilon$. By combining this result with the robust mean

estimation result of [DHL19], we obtain the following result for learning an arbitrary Gaussian, in total variation distance:

**Corollary 2.3.** *Let $D = \mathcal{N}(\mu, \Sigma)$ be an arbitrary Gaussian, and let $\kappa$ be the condition number of $\Sigma$. Let $\varepsilon < c$ for some universal constant $c$, and let $S$ be an $\varepsilon$-corrupted set of samples from $D$ of size $N = \widetilde{\Omega}(d^2/\varepsilon^2)$. Then, there is an algorithm which takes $S, \varepsilon$ and outputs $\widehat{\mu}, \widehat{\Sigma}$ so that $\mathrm{d_{TV}}(D, \mathcal{N}(\widehat{\mu}, \widehat{\Sigma})) \leq O(\varepsilon \log 1/\varepsilon)$. Moreover, the algorithm runs in time $\widetilde{O}(T(N, d) \log \kappa)$.*

Second, note that the runtime of our algorithm, up to poly-logarithmic factors, and the logarithmic dependence on $\kappa$, matches that of the best known non-robust algorithm. This runtime strictly improves upon the runtime of the algorithm in [CDGW19] with the same guarantee. The authors of [CDGW19] also give another algorithm which avoids the $\log \kappa$ dependence in the runtime, but only guarantees that $\|\Sigma - \widehat{\Sigma}\|_F \leq O(\varepsilon \log 1/\varepsilon)\|\Sigma\|$. Note that this weaker guarantee does not yield any meaningful statistical guarantees.

Third, it is well-known (see e.g. [CZZ$^+$10]) that, even without corruptions, $\Omega(d^2/\varepsilon^2)$ samples are necessary to learn the covariance to Mahalanobis distance $O(\varepsilon)$. Thus, our algorithm is nearly sample optimal for this problem. Moreover, in the presence of corruptions, it is likely that the $\Omega(d^2)$ in the sample complexity is unavoidable for efficient algorithms, even if we relax the problem and ask for weaker guarantees, such as spectral approximation, or approximation in PSD ordering [DKS17].

Finally, our error guarantee of $O(\varepsilon \log 1/\varepsilon)$ is off from the optimal error of $O(\varepsilon)$ by a logarithmic factor. However, this is also likely unavoidable for efficient algorithms in this strong model of corruption [DKS17]. It is known that in slightly weaker notions of corruption such as Huber's contamination model, this can be improved in quasi-polynomial time [DKK$^+$18]. It is an interesting open question whether or not this can be achieved in polynomial time.

## 3 Mathematical Preliminaries

### 3.1 Notation

For two functions $f, g$, we say $f = \widetilde{O}(g)$ if $f = O(g \log^c g)$ for some universal constant $c > 0$. We similarly define $\widetilde{\Omega}$ and $\widetilde{\Theta}$. For vectors $v \in \mathbb{R}^d$, we let $\|\cdot\|_2$ denote the usual $\ell_2$ norm, and $\langle \cdot, \cdot \rangle$ denote the usual inner product between vectors.

For any $N$, we let $\Gamma_N = \{w \in \mathbb{R}^N : 0 \leq w_i \leq 1, \sum w_i \leq 1\}$ denote the set of vectors which are valid reweightings of $N$. Note that we allow for these weightings to sum up to less than 1. For any $w \in \Gamma_N$, we let $|w| = \sum w_i$ be its mass. Moreover, given a set of points $Z_1, \ldots, Z_N$, and $w \in \Gamma_N$, let $\mu(w) = \frac{1}{|w|} \sum w_i Z_i$, and $M(w) = \frac{1}{|w|} \sum w_i (Z_i - \mu(w))(Z_i - \mu(w))^\top$ denote the empirical mean and variance of this set of points with the weighting given by $w$, respectively.

For matrices $A, M \in \mathbb{R}^{d \times d}$ we let $\|M\|$ denote its spectral norm, we let $\|M\|_F$ denote its Frobenius norm, and we let $\langle M, A \rangle = \mathrm{tr}(M^\top A)$ denote the trace inner product between matrices. For any symmetric matrix $A \in \mathbb{R}^{d \times d}$, let $\exp(A)$ denote the usual matrix exponential of $A$. Finally, for scalars $x, y \in \mathbb{R}$, and any $\alpha > 0$, we say that $x \approx_\alpha y$ if $\frac{1}{1+\alpha} x \leq y \leq (1 + \alpha) x$.

### 3.2 Naive Pruning

As a simple but useful preprocessing step, we will need to be able to remove points that are "obviously" outliers. It's known that there exists a randomized algorithm achieving this with nearly-linear many $\ell_2$ distance queries:

**Lemma 3.1** (folklore). *There is an algorithm* NAIVEPRUNE *with the following guarantees. Let $0 < \varepsilon < 1/2$. Let $S \in \mathbb{R}^m$ be a set of $n$ points so that there exists a ball $B$ with radius $r$ and a subset $S' \subseteq S$ so that $|S'| \geq (1 - \varepsilon)n$ and $S' \subset B$. Then, with probability $1 - \delta$,* NAIVEPRUNE$(S, r, \delta)$ *outputs a set of points $T \subseteq S$ so that $S' \subseteq T$, and $T$ in contained in a ball of radius $4r$. Moreover, if all points $Z_i \in S$ are of the form $Z_i = X_i \otimes X_i$ for $X_i \in \mathbb{R}^d$, then* NAIVEPRUNE$(S, r, \delta)$ *can be implemented in $\widetilde{O}(T(N, d) \log(1/\delta))$ time.*

For completeness, we prove this lemma in Appendix A.

### 3.3 Quantum Entropy Score Filtering

A crucial primitive that we will use throughout this paper is the quantum entropy scoring-based filters of [DHL19]. To instantiate the guarantees of these algorithms, we require two ingredients: (1) regularity conditions under which the algorithm is guaranteed to work, and (2) score oracles (or approximate score oracles), which compute the scores which the algorithm will use to downweight outliers. In this section, we will define these concepts, and state the guarantees that quantum entropy scoring achieves. The reader is referred to [DHL19] for more details on the actual implementation of the filtering algorithms.

#### 3.3.1 Regularity Condition

The filtering algorithms can be shown to work under a set of general regularity conditions imposed on the original set of uncorrupted data points. Formally:

**Definition 3.2.** *Let $D$ be a distribution over $\mathbb{R}^m$ with unknown mean $\mu$ and covariance $\Sigma \preceq \sigma^2 I$. We say a set of points $S \subseteq \mathbb{R}^m$ is $(\varepsilon, \gamma_1, \gamma_2, \beta_1, \beta_2)$-good with respect to $D$ if there exists universal constants $C_1, C_2$ so that the following inequalities are satisfied:*

- *$\|\mu(S) - \mu\|_2 \leq \sigma\gamma_1$ and $\|\frac{1}{|S|} \sum_{i \in S}(X_i - \mu(S))(X_i - \mu(S))^\top - \Sigma\| \leq \sigma^2 \gamma_2$.*

- *For any subset $T \subset S$ so that $|T| = 2\varepsilon|S|$, we have*

$$\left\| \frac{1}{|T|} \sum_{i \in T} X_i - \mu \right\|_2 \leq \beta_1, \ and \ \left\| \frac{1}{|T|} \sum_{i \in T}(X_i - \mu(S))(X_i - \mu(S))^\top - \Sigma \right\| \leq \beta_2 \ .$$

*If a set of points is $(\varepsilon, \gamma_1, \gamma_2, \infty, \infty)$-good with respect to $D$, we say that it is $(\gamma_1, \gamma_2)$-good with respect to $D$ (Note that when $\beta_1 = \beta_2 = \infty$, the condition now becomes independent of $\varepsilon$).*

Additionally, we will say that a set of points $S$ is $(\varepsilon, \gamma_1, \gamma_2, \beta_1, \beta_2)$-corrupted good (resp. $(\varepsilon, \gamma_1, \gamma_2)$-corrupted good) with respect to $D$ if it can be written as $S = S_g \cup S_b \setminus S_r$, where $S_g$ is $(\varepsilon, \gamma_1, \gamma_2, \beta_1, \beta_2)$-good (resp. $(\varepsilon, \gamma_1, \gamma_2)$-good) with respect to $D$, and we have $|S_r| = |S_b| \leq \varepsilon|S|$.

Intuitively, a set of points is corrupted good if it is an $\varepsilon$-corrupted of a good set of points.

#### 3.3.2 Score oracles and variants thereof

The idealized score oracle takes as input an integer $t > 0$, a set of points $Z_1, \ldots, Z_N$, and a sequence of weight vectors $w_0, \ldots, w_{t-1} \in \Gamma_N$, and outputs $\lambda = \|M(w_0) - I\|_2$ as well as $\tau_{t,i}$, for $i = 1, \ldots, N$, where $\tau_{t,i}$ is the quantum entropy score (QUE score), and is defined to be:

$$\tau_{t,i} = (Z_i - \mu(w_t))^\top U_t (Z_i - \mu(w_t)) \ , \tag{2}$$

where

$$U_t = \frac{\exp\left(\alpha \sum_{i=0}^{t-1} M(w_i)\right)}{\operatorname{tr} \exp\left(\alpha \sum_{i=0}^{t-1} M(w_i)\right)} \ .$$

Here $\alpha > 0$ is a parameter which will be tuned by the QUE-score filtering algorithm, and we will always choose $\alpha$ so that

$$\left\| \alpha \sum_{i=0}^{t-1} M(w_i) \right\| \leq O(t) \ . \tag{3}$$

However, computing this exact score oracle is quite inefficient, so for runtime purposes, we will instead typically work with approximate score oracles.

An approximate score oracle, which we will denote $\mathcal{O}_{\text{approx}}$, is any algorithm, which given input as above, instead outputs $\tilde{\lambda}$ and $\tilde{\tau}_{t,i}$ for $i = 1, \ldots, N$ so that $\tilde{\lambda} \approx_{0.1} \|M(w_0) - I\|$, and $\tilde{\tau}_{t,i} \approx_{0.1} \tau_{t,i}$ for all $i = 1, \ldots, N$, where $\tau_{t,i}$ is defined as in (2).

Note that this is slightly different from the definition of the score oracle in [DHL19], as there we do not require that we also output the spectral norm of $M(w_0) - I$. This is because, in the original setting

of [DHL19], this computation could be straightforwardly done via power method. However, our setting is more complicated and doing so requires more work in our setting, and so it will be useful to encapsulate this computation into the definition of the score oracle. Another slight difference is that here we ask the oracle to output a multiplicative approximation to the spectral norm of $M(w_0) - I$, but in some settings in [DHL19], we ask for a multiplicative approximation of $\|M(w_0)\|$. However, it is easily verified that in the settings we care about, we will always have $\|M(w_0)\| \geq 0.9$, and thus a multiplicative approximation of $\|M(w_0) - I\|$ will always be sufficient for our purposes.

In addition, we say that the score oracle is an approximate augmented score oracle, denoted $\mathscr{O}_{\mathrm{aug}}$, if in addition, it outputs $\tilde{q}_t$, which is defined to be any value satisfying:

$$|\tilde{q}_t - q_t| \leq 0.1 q_t + 0.05 \|M(w_t) - I\|, \text{ where } q_t = \langle M(w_t) - I, U_t \rangle . \tag{4}$$

### 3.3.3 Guarantees of QUE score filtering

Given these two definitions, we can now state the guarantees of the QUE scoring algorithms. The first theorem allows us to achieve a somewhat coarse error guarantee, under $(\gamma_1, \gamma_2)$-goodness:

**Lemma 3.3** (Theorem 2.1 in [DHL19]). *Let $D$ be a distribution on $\mathbb{R}^m$ with unknown mean $\mu$ and covariance $\Sigma \preceq \sigma^2 I$, for $\sigma \geq 1$. Let $0 < \varepsilon < c$ for some universal constant $c$. Let $S$ be an $(\varepsilon, O(\sqrt{\varepsilon}), O(1))$-corrupted good set of points with respect to $D$. Suppose further that $\|X\|_2 \leq R$ for all $X \in S$. Let $\mathscr{O}_{\mathrm{approx}}$ be an approximate score oracle for $S$. Then, there is an algorithm which outputs an vector $\hat{\mu} \in \mathbb{R}^m$ such that $\|\hat{\mu} - \mu\| \leq O(\sigma\sqrt{\varepsilon})$. Moreover, this algorithm requires $O(\log(mn) \log m)$ calls to $\mathscr{O}_{\mathrm{approx}}$ with input $t \leq O(\log m)$ and $\alpha$ satisfying Equation (3), and requires $\widetilde{O}(n \log(R/\sigma))$ additional computation.*

The second theorem allows us to refine our error estimate in the second phase, under a stronger assumption on the goodness of the corrupted set, and with access to an augmented score oracle:

**Lemma 3.4** (Theorem 4.7 in [DHL19]). *Let $D$ be a distribution on $\mathbb{R}^m$ with covariance $\Sigma$ satisfying $\|\Sigma\| \leq O(1)$. Let $\varepsilon < c$, where $c$ is a universal constant where $c$ is a universal constant, let $\gamma_1, \gamma_2.\beta_1, \beta_2 > 0$. Let $S$ be a $(\varepsilon, \gamma_1, \gamma_2.\beta_1, \beta_2)$-corrupted good set with respect to $D$. Suppose further that $\|X\|_2 \leq R$ for all $X \in S$. Let $\mathscr{O}_{\mathrm{aug}}$ be an approximate augmented score oracle for $S$. Then, there is an algorithm which outputs $\hat{\mu}$ so that*

$$\|\hat{\mu} - \mu\|_2 \leq O\left(\gamma_1 + \varepsilon\sqrt{\log 1/\varepsilon} + \sqrt{\varepsilon\xi}\right) ,$$

*where*

$$\xi = \xi(\varepsilon, \gamma_1, \gamma_2, \beta_1, \beta_2) = \gamma_2 + 2\gamma_1^2 + 4\varepsilon^2\beta_1^2 + 2\varepsilon\beta_2 + O(\varepsilon \log 1/\varepsilon) . \tag{5}$$

*Moreover, the algorithm requires $O(\log d \log R)$ calls to $\mathscr{O}_{\mathrm{aug}}$ with input $t \leq O(\log d)$ and $\alpha$ satisfying Equation (3), and requires $\widetilde{O}(n \log R)$ additional computation.*

## 4 Proof of Theorem 2

In this section, we prove our main theorem modulo a number of key technical lemmata, whose proofs we defer to later sections. We restate the main theorem here for convenience.

**Theorem 2** (Main Theorem). *Let $D = \mathcal{N}(\mathbf{0}_d, \Sigma)$ be a zero-mean unknown covariance multivariate Gaussian over $\mathbb{R}^d$. Let $\kappa$ be the condition number of $\Sigma$. Let $0 < \varepsilon < c$, where $c$ is a universal constant. Let $S$ be a $\varepsilon$-corrupted set of samples from $D$ of size $N = \widetilde{\Omega}(d^2/\varepsilon^2)$. Algorithm 1 that runs in time $\widetilde{O}(T(N, d) \log \kappa)$ takes $S$ and $\varepsilon$, and outputs a $\widehat{\Sigma}$ so that with probability at least $0.99$, we have $\|\Sigma^{-1/2}\widehat{\Sigma}\Sigma^{-1/2} - I\|_F \leq O(\varepsilon \log(1/\varepsilon))$.*

We do this in a couple of steps. The first is a reduction from robust covariance estimation to robust mean estimation. As observed in [CDGW19], when $X \sim \mathcal{N}(\mathbf{0}_d, \Sigma)$, we have $\mathbb{E}[XX^\top] = \Sigma$, so basically the covariance estimation problem is equivalent to estimating the mean of the tensor product $X \otimes X$. One of the main difficulties for adapting the existing algorithms for robust mean estimation is that those algorithms either assume that the distribution is isotropic or has bounded covariance. However, the covariance of $X \otimes X$ corresponds to the fourth moments of $X$, which can depend in a complicated way on the (unknown) $\Sigma$. To solve this problem, we adapt the iterative refinement

technique from [CDGW19]. Basically, given an upper bound $\Sigma_t \succeq \Sigma$, we can use this upper bound in a robust mean estimation sub-routine to compute a more accurate upper bound $\Sigma_{t+1}$, and recurse.

In prior work of [CDGW19], this refinement step was done using a black-box call to a packing SDP. Our goal is to show that this call can be replaced by a call to a QUE-score filtering algorithm, as this is what will allow us to avoid the $\mathrm{poly}(1/\varepsilon)$ dependence in the runtime. The main technical work will be to demonstrate that the data has sufficient regularity conditions so that QUE-scoring will succeed, and that we can construct the appropriate approximate score oracles.

## 4.1 Deterministic Regularity Conditions

We first require the following definition:

**Definition 4.1.** *For any positive definite $\Sigma$, let $D_\Sigma$ denote the distribution of $Y = X \otimes X$, where $X \sim \mathcal{N}(0, \Sigma)$.*

Throughout the remainder of the proof, we will condition on the following, deterministic regularity condition on the dataset $S$:

**Assumption 4.2.** *The dataset $S$ can be written as $S = S_g \cup S_b \setminus S_r$, where $|S_b| = |S_r| = \varepsilon N$, for some $\varepsilon$ sufficiently small, and $S_g = \{X_1, \ldots, X_N\}$, where $X_i = \Sigma^{1/2} \bar{X}_i$, for $i = 1, \ldots, N$, and the set $\{\bar{X}_1 \otimes \bar{X}_1, \ldots, \bar{X}_N \otimes \bar{X}_N\}$ is*

$$(\varepsilon, O(\varepsilon\sqrt{\log 1/\varepsilon}), O(\varepsilon \cdot \log^2 1/\varepsilon), O(\log 1/\varepsilon), O(\log^2 1/\varepsilon))\text{-good with respect to } D_I .$$

*Moreover, all of the $\bar{X}_i$ satisfy $\|X_i\|_2^2 \leq O(d \log N)$.*

In Appendix B, we show the following lemma:

**Lemma 4.3.** *Let $S$ be an $\varepsilon$-corrupted set of samples from $\mathcal{N}(0, \Sigma)$ of size $N = \widetilde{\Omega}\left(d^2/\varepsilon^2\right)$. Then, with probability at least $1 - d^{-3}$, the set $S$ satisfies Assumption 4.2.*

A key consequence of Assumption 4.2 will be that the set of points $S_g$ will satisfy strong goodness conditions, even after rotations are applied. Specifically:

**Lemma 4.4.** *Let $S$ and $S_g$ be as in Assumption 4.2, and let $\Sigma$ be positive definite. Then if we let $Z_i = (\Sigma^{1/2} X_i) \otimes (\Sigma^{1/2} X_i)$, then the set $\{Z_1, \ldots, Z_N\}$ is $(\varepsilon, O(\sqrt{\varepsilon}), O(1))$-good with respect to $D_\Sigma$.*

*In addition, if $\Sigma$ satisfies $\|\Sigma - I\| \leq \xi$ for some $\xi < 1$, then the set $\{Z_1, \ldots, Z_N\}$ is*

$$(\varepsilon, O(\varepsilon\sqrt{\log 1/\varepsilon}), O(\varepsilon \cdot \log^2 1/\varepsilon) + 6\zeta, O(\log 1/\varepsilon), O(\log^2 1/\varepsilon) + 6\zeta)\text{-good with respect to } D_I .$$

## 4.2 Algorithm Description

We now describe the crucial subroutines which will allow us to achieve Theorem 2. We will use two phases of iterative refinement steps (Appendices D.2 and D.3), which we will describe and analyze separately. The first phase will allow us to estimate the covariance relatively coarsely. Then, the second phase, we will use the fact that if our estimation $\Sigma_t$ is already close to $\Sigma$, then $Y_i = \Sigma_t^{-1/2} X_i$ has covariance close to the identity matrix. This allows us to invoke the stronger QUE scoring algorithm, which allows us to refine the estimate all the way down to $O(\varepsilon \log 1/\varepsilon)$.

First, we need to get a rough estimation of $\Sigma$ as the initial point, so that we can apply the iterative refinement steps. We invoke the following lemma:

**Lemma 4.5** (Lemma 3.1 of [CDGW19]). *Consider the same setting as in Theorem 2. We can compute a matrix $\Sigma_0$ in $\widetilde{O}(T(N, d))$ such that, with high probability, $\Sigma \preceq \Sigma_0 \preceq (\kappa \, \mathrm{poly}(d))\Sigma$ and $\|\Sigma_0\| \leq \mathrm{poly}(d)\|\Sigma\|$.*

We first give an algorithm FIRSTPHASE, which, given an upper bound on $\Sigma$, outputs a relatively coarse approximation to $\Sigma$ :

**Theorem 4.6** (First Phase). *Let $S$ be a set of points satisfying Assumption 4.2. Moreover, let $\Sigma_t \in \mathbb{R}^{d \times d}$ be such that $\Sigma \preceq \Sigma_t$. Then there is an algorithm FIRSTPHASE, which given $S$ and $\Sigma_t$,*

---
**Algorithm 1** Robust Covariance Estimation
---
1: **Input:** $S = \{X_1, X_2, \ldots, X_n\}, \varepsilon$
2: $T_1 \leftarrow O(\log \kappa + \log d), T_2 \leftarrow T_1 + O(\log \log(1/\varepsilon))$
3: Compute an initial upper bound $\Sigma_0$ use Lemma 4.5.
4: **for** $t = 1, \ldots, T_1 - 1$ **do**
5:     $\Sigma_{t+1} \leftarrow \text{FIRSTPHASE}(S, \Sigma_t)$                                       ▷ Algorithm 2
6: **end for**
7: $\zeta_{T_1} \leftarrow O(\sqrt{\varepsilon})$
8: **for** $t = T_1, \ldots, T_2$ **do**
9:     $\widehat{\Sigma}_{t+1}, \Sigma_t, \zeta_{t+1} \leftarrow \text{SECONDPHASE}(S, \Sigma_{t+1}, \zeta_t)$                     ▷ Algorithm 3
10: **end for**
11: **return** $\widehat{\Sigma}_{T_2+1}$
---

*runs in time* $\widetilde{O}(T(N, d) \log \log \kappa)$ *and outputs a new upper bound matrix* $\Sigma_{t+1}$ *and a approximate covariance matrix* $\widehat{\Sigma}$ *such that, with probability* $1 - 1/\operatorname{poly}(d, \log \kappa)$,

$$\Sigma \preceq \Sigma_{t+1} \preceq \Sigma + O(\sqrt{\varepsilon})\Sigma_t \,, \qquad and \qquad \|\widehat{\Sigma} - \Sigma\|_F \leq O(\sqrt{\varepsilon})\|\Sigma_t\| \,.$$

In the second phase, since we already have a somewhat accurate estimation of $\Sigma$, we show that we can use this to get a matrix with $O(\varepsilon \log 1/\varepsilon)$ error.

**Theorem 4.7** (Second Phase). *Let $S$ be a set of points satisfying Assumption 4.2. Let $0 < \zeta < \zeta_0$ for some universal constant $\zeta_0$. Given $\zeta_t$ and $\Sigma_t$ where $\Sigma \preceq \Sigma_t \preceq (1 + \zeta_t)\Sigma$ as input, Algorithm 3 runs in time $\widetilde{O}(T(N, d))$ and outputs a new upper bound matrix $\Sigma_{t+1}$ and a approximate covariance matrix $\widehat{\Sigma}$ such that, with probability $1 - 1/Nd$, for $\zeta_{t+1} = O(\sqrt{\varepsilon \zeta_t} + \varepsilon \log 1/\varepsilon)$, we have*

$$\Sigma \preceq \Sigma_{t+1} \preceq \Sigma + \zeta_{t+1}\Sigma_t \,, \qquad and \qquad \|\Sigma^{-1/2}\widehat{\Sigma}\Sigma^{-1/2} - I\|_F \leq \zeta_{t+1} \,.$$

We defer the proof of Theorem 4.6 to Appendix D.2 and the proof of Theorem 4.7 to Appendix D.3. Now, assuming Lemma 4.5 and Theorems 4.6 and 4.7, we are ready to prove Theorem 2.

*Proof of Theorem 2.* By Lemma 4.3, our dataset satisfies Assumption 4.2 with probability $1 - 1/d^3$. Condition on this event holding for the remainder of the proof. By Lemma 4.5, we have $\Sigma \preceq \Sigma_0 \preceq (\kappa \operatorname{poly}(d))\Sigma$. For any given covariance upperbound matrix $\Sigma_t$, we use FIRSTPHASE to get a more accurate upperbound $\Sigma_{t+1} \preceq \Sigma + O(\sqrt{\varepsilon})\Sigma_t$, then after $O(\log \kappa + \log d)$ iterations, we have $\Sigma_{T_1} \preceq (1 + O(\sqrt{\varepsilon}))\Sigma_{t+1}$.

Since we already have a good estimation on covariance where $\zeta_{T_1} = O(\sqrt{\varepsilon})$, we use SECONDPHASE to obtain a better estimation. By Theorem 4.7, we know that $\zeta_{t+1} = O(\sqrt{\varepsilon \zeta_t} + \varepsilon \log(1/\varepsilon))$, then after $\log \log(1/\varepsilon)$ iterations, we have $\zeta_{T_2} \leq O(\varepsilon \log(1/\varepsilon))$. Then, by the guarantee on $\widehat{\Sigma}$, we have

$$\|\Sigma^{-1/2}\widehat{\Sigma}_{T_2}\Sigma^{-1/2} - I\|_F = O(\sqrt{\varepsilon \zeta_{T_2}}) = O(\varepsilon \log(1/\varepsilon)).$$

Now, we consider the probability of success. By Lemma 4.5, we compute $\Sigma_0$ with probability $1 - \frac{1}{d}$. In the first phase, each iteration succeed with probability at least $1 - \frac{1}{\operatorname{poly}(d, \log k)}$ by Theorem 4.6, since we have $O(\log d + \log \kappa)$ iterations in the first phase, then first phase succeed with probability $1 - \frac{1}{\operatorname{poly}(d)}$. Similarly, by Theorem 4.7, each iteration succeed with probability $1 - \frac{1}{Nd}$ and we runs this for $\log \log(1/\varepsilon)$ iterations, since $N = \Omega(d^2/\varepsilon^2)$, then all the iterations of second phase succeed with probability at least $1 - \frac{1}{d}$. By union bound over all failure probability, we conclude that Algorithm 1 succeed with probability at least $1 - O(1/d) \geq 0.99$.

For the running time, note that we can compute $\Sigma_0$ in $O(T(N, d))$ time and we run $O(\log \kappa + \log d + \log \log(1/\varepsilon))$ iterations in total. In each iteration, we either call FIRSTPHASE or SECONDPHASE, where both of them have runtime $\widetilde{O}(T(N, d) \log \log \kappa)$. Thus, the overall runtime is $\widetilde{O}(T(N, d) \log \kappa)$. $\square$

## Broader Impact

Moving forward, it is imperative that machine learning systems cannot be gamed by malicious entities. This work builds upon a growing literature of principled algorithms for robust statistics, which are methods for defending against data poisoning attacks, where a training set may be tampered with by an adversary who wishes to change the behavior of the algorithm. For instance, such defenses are important in where the training data is crowdsourced, such as in federated learning, where we cannot fully trust the training data. In such settings, if the defense fails, attackers can completely invalidate the output of the model. That is why we believe it is critical to develop principled defenses, with provable worst-case guarantees, as we do here. With such defenses, we know that this worst-case behavior cannot happen.

The algorithms developed here are also useful for exploratory data analysis, as demonstrated in [DKK+17]. Most real-world high-dimensional datasets are inherently very noisy, and this noise can disguise interesting patterns from data analysts. These methods can be used in exploratory data analysis to remove this noise, and to recover these phenomena.

We do not believe that this method leverages any biases in the data. Our generative model, as stated in the introduction, is very simple, and does not introduce any biases in this problem.

## Acknowledgments and Disclosure of Funding

Guanghao Ye was supported in part by NSF Awards CCF-1740551, CCF-1749609, and DMS-1839116.

## Footnotes

[1]Throughout this paper, we say $f = \widetilde{O}(g)$ if $f = O(g \log^c g)$ for some universal constant $c > 0$.

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
