[Supplementary Material]

## A Proof of Lemma 3.1

*Proof of Lemma 3.1.* The algorithm is straightforward: choose a random point in $S$, and check if strictly more than $n/2$ points lie within a ball of radius $2r$ around this point. If so, include all points with distance at most $4r$ from this point. Note that we cannot calculate the $\ell_2$ distance directly, instead, using JL-lemma, we project all points onto $\mathbb{R}^{O(\log d)}$. If not, repeat, and run for $O(\log 1/\delta)$ iterations.

Similar to proof of Lemma E.5, let $J \in \mathbb{R}^{r \times d^2}$ matrix whose each entries are i.i.d. entries from $\mathcal{N}(0, 1/r)$ where $r = O(\log d)$. Note that $Zv = (Y \operatorname{diag}(v) Y^\top)^\flat$, then we can compute $J \cdot Z$ using fact rectangular matrix multiplication by multiply each row of $J$ to $Z$. Then, this takes $\widetilde{O}(T(N, d))$ time. Note that for each iteration, we compute $O(N)$ many $\ell_2$ distance, which takes time $\widetilde{O}(N)$. Thus, the total time is $\widetilde{O}(T(N, d) + N \log(1/\delta)) = \widetilde{O}(T(N, d) \log(1/\delta))$.

By the triangle inequality, if we ever randomly select a point from $S'$, then we terminate, and in this case it is easy to see that the output satisfies the desired property. Thus, it is easy to see that the probability we have not terminated after $t$ iterations is at most $2^{-t}$. Suppose we have terminated. Then in that iteration, we selected a point $X \in S$ that has distance at most $2r$ to more than $n/2$ other points in $S$. This implies that it has distance at most $2r$ to some point in $S'$. By triangle inequality, this implies that all points in $S'$ are at distance at most $4r$ from $X$, and so the output in this iteration must satisfy the claims of the Lemma. $\qquad\square$

## B Proof of Lemma 4.3

Before we prove Lemma 4.3, we need the following preliminaries.

**Lemma B.1** (Hanson-Wright). *Let $X_1, X_2, \ldots X_N$ be i.i.d. random vectors in $\mathbb{R}^d$ where $X_i \sim \mathcal{N}(0, \Sigma)$ and $\Sigma \preceq I$. Let $U \in \mathbb{R}^{d \times d}$ and $U \succeq 0$ and $\|U\|_F = 1$. Then, there exists a universal constant $C$ so that for all $T > 0$, we have*

$$\Pr\left[\left|\frac{1}{N}\sum_{i=1}^{n} \operatorname{tr}(X_i X_i^\top U) - \operatorname{tr}(U)\right| > T\right] \leq 2\exp(-CN\min(T, T^2)).$$

**Corollary B.2.** *Under the same setting as Lemma B.1, let $Z_i = X_i \otimes X_i$ and $v \in \mathbb{R}^{d^2}$ be an arbitrary unit vector. Then, there exists a universal constant $C > 0$ so that for all $T > 0$, we have*

$$\Pr\left[\left|\frac{1}{N}\sum_{i=1}^{n}\langle v, Z_i\rangle - \mathbb{E}[\langle v, Z\rangle]\right| > T\right] \leq 2\exp(-CN\min(T, T^2)).$$

*Proof.* This follows by letting the $v$ in the statement be the flattening of $U$ in Lemma B.1. $\qquad\square$

Using a standard $\varepsilon$-net argument (see e.g. [Ver10]), we get the following concentration bounds for the empirical mean.

**Lemma B.3.** *Under the same setting as Lemma B.1, let $Z_i = X_i \otimes X_i$ there exist universal constants $A, C > 0$ so that for all $T > 0$, we have*

$$\Pr\left[\left\|\frac{1}{N}\sum_{i=1}^{n} Z_i - \mu_Z\right\|_2 > T\right] \leq 2\exp(Ad^2 - CN\min(T, T^2)).$$

Now, we are ready to prove Lemma 4.3.

*Proof of Lemma 4.3.* The parameter of $\gamma_1$ directly follows from Lemma B.3 by choosing $T = \varepsilon\sqrt{\log(1/\varepsilon)}$ and $N = \Omega(\frac{d^2}{\varepsilon^2 \log(1/\varepsilon)})$.

For $\beta_1$, we apply Lemma B.3 for any fixed set $S \subset [N]$ of size $2\varepsilon N$, we have

$$\Pr\left[\left\|\frac{1}{|S|}\sum_{i \in S} Z_i - \mu_Z\right\|_2 > T\right] \leq 2\exp(Ad^2 - C\varepsilon N\min(T^2, T)).$$

Taking the union bound over all subsets of size $2\varepsilon N$, we get

$$\Pr\left[\exists S : |S| = 2\varepsilon N \text{ and } \left\|\frac{1}{|S|}\sum_{i \in S} Z_i - \mu_Z\right\|_2 > T\right]$$

$$\leq 2\exp(Ad^2 + \log\binom{2\varepsilon N}{N} - C\varepsilon N \min(T^2, T))$$

$$\leq 2\exp(Ad^2 + O(N \cdot \varepsilon \log 1/\varepsilon) - C\varepsilon N \min(T^2, T)).$$

By choose $T = O(\log(1/\varepsilon))$ and $N = \Omega(\frac{d^2}{\varepsilon^2 \log(1/\varepsilon)})$, we have the probability above is less than $O(d^{-3})$.

For proof of $\gamma_2$ and $\beta_2$, see Theorem 4.13 of [DKK$^+$16] and Proposition A.28 of [DKK$^+$17]. □

## C   Proof of Lemma 4.4

Before we prove this lemma, we require the following pair of technical lemmata. The first is a standard fact about the covariance of $X \otimes X$ for $X$ Gaussian.

**Fact C.1** (see e.g. [CDGW19]). *Let $X \sim \mathcal{N}(0, \Sigma)$. Then the covariance of $X \otimes X$ is $2\Sigma \otimes \Sigma$.*

This implies:

**Lemma C.2.** *Let $X \sim \mathcal{N}(\mathbf{0}, \Sigma)$ and $Z = X \otimes X$. Let $\Sigma_Z \in \mathbb{R}^{d^2 \times d^2}$ be the covariance matrix of $Z$. We have:*
*1. If $\Sigma \preceq I$, then $\Sigma_Z \preceq 2I$.*
*2. If $\|\Sigma - I\| \leq \zeta$ for $0 \leq \zeta < 1$, then $\|\Sigma_Z - 2I\| \leq 6\zeta$.*

*Proof.* The first claim follows directly from Fact C.1, as $\|\Sigma_Z\| = 2\|\Sigma\| \leq 2$. To prove the second statement, note that if $\lambda_1, \ldots, \lambda_d$ are the eigenvalues of $\Sigma$, the assumption implies that $|\lambda_i - 1| \leq \zeta$ for all $i$, and also that $\lambda_i < 2$ for all $i$. But all the eigenvalues of $\Sigma_Z$ are given by $2\lambda_i\lambda_j$ for $i, j \in [d]$, and $|2\lambda_i\lambda_j - 2| \leq 2(\zeta^2 + |\lambda_i|\zeta + |\lambda_j|\zeta) \leq 6\zeta$. This proves the claim. □

*Proof of Lemma 4.4.* The first claim follows because $(\gamma_1, \gamma_2)$-goodness is affine invariant. We now turn our attention to the second claim. First, we show that the $\gamma_1$ and $\beta_1$ parameters are changed by at most a constant multiplicative factor. Since $X_i = \Sigma^{-1/2}\overline{X}_i$, then we have

$$Z_i = (\Sigma^{1/2} \otimes \Sigma^{1/2})(\overline{X}_i \otimes \overline{X}_i) = (\Sigma^{1/2} \otimes \Sigma^{1/2})\overline{Z}_i.$$

Then, we have

$$\gamma_1(Z_i) = \|\mu(Z_i) - \mu_{Z_i}\|_2 = \left\|(\Sigma^{1/2} \otimes \Sigma^{1/2})(\mu(\overline{Z}_i) - \mu_{\overline{Z}_i})\right\|_2 \leq O(\varepsilon \log 1/\varepsilon),$$

where the last step follows by $\|A \otimes B\| \leq \|A\|\|B\|$ and $\|\Sigma\| \leq 2$. Similarly, we have that the $\beta_1$ parameter increases by at most a constant multiplicative factor.

Now, we consider the second moment parameters $\gamma_2$ and $\beta_2$. Note that we have $Z_i Z_i^\top = (\Sigma \otimes \Sigma)^{1/2} \overline{Z}_i \overline{Z}_i^\top (\Sigma \otimes \Sigma)^{1/2}$. By the goodness of $\overline{Z}_i$, then we have

$$\left\|\frac{1}{|S|}\sum_{i \in S}(Z_i - \mu(S))(Z_i - \mu(S))^\top - 2(\Sigma \otimes \Sigma)\right\| \leq \|\Sigma\|^2 \cdot O(\varepsilon\sqrt{\log 1/\varepsilon}) = O(\varepsilon\sqrt{\log 1/\varepsilon}).$$

Then, by Lemma C.2, we have $\|\Sigma \otimes \Sigma - 2I\| \leq 6\zeta$, so by, we have

$$\left\|\frac{1}{|S|}\sum_{i \in S}(Z_i - \mu(S))(Z_i - \mu(S))^\top - 2I\right\| \leq O(\varepsilon\sqrt{\log 1/\varepsilon}) + 6\zeta,$$

as claimed. The bound on the $\beta_2$ parameter is identical, and omitted. □

# D Proof of Theorem 4.6 and Theorem 4.7

## D.1 Approximate Score Oracles for Tensor Inputs

A key algorithmic ingredient to implementing both Theorem 4.6 and Theorem 4.7 will be the following. We will be given access to a set of points $X_1, \ldots, X_N$, and we will need approximate augmented score oracles for the tensored set of points $X_1 \otimes X_1, \ldots, X_N \otimes X_N$. Note that we cannot even afford to write down the tensored versions of the $X_i$ in the desired runtime. Despite this, we show that we can construct these approximate augmented score oracles very efficiently:

**Theorem D.1.** *Let $\delta > 0$. Let $X_1, \ldots, X_N \in \mathbb{R}^d$, and let $Z_i = X_i \otimes X_i$ for all $i = 1, \ldots, N$. Let $t > 0$, and let $w_1, \ldots, w_t \in \Gamma_N$. Let $\alpha$ be such that $\alpha$ satisfies Equation (3). Then, there is an algorithm* APPROXIMATESCORE *which takes as input $\alpha, \delta, \{X_1, \ldots, X_n\}$, and $w_1, \ldots, w_t$, which runs in time $\widetilde{O}(t^2 \cdot T(N, d) \log 1/\delta)$, and with probability $1 - \delta$, is an approximate augmented score oracle for $\{Z_1, \ldots, Z_N\}$ with weights $w_1, \ldots, w_t$.*

We defer the proof to Section E.

## D.2 Getting $O(\sqrt{\varepsilon})$ error

In this section, we describe and analyze the routine FIRSTPHASE, which achieves a coarse estimate of the true covariance. We restate the theorem below for convenience.

**Theorem 4.6** (First Phase). *Let $S$ be a set of points satisfying Assumption 4.2. Moreover, let $\Sigma_t \in \mathbb{R}^{d \times d}$ be such that $\Sigma \preceq \Sigma_t$. Then there is an algorithm* FIRSTPHASE, *which given $S$ and $\Sigma_t$, runs in time $\widetilde{O}(T(N, d) \log \log \kappa)$ and outputs a new upper bound matrix $\Sigma_{t+1}$ and a approximate covariance matrix $\widehat{\Sigma}$ such that, with probability $1 - 1/\operatorname{poly}(d, \log \kappa)$,*

$$\Sigma \preceq \Sigma_{t+1} \preceq \Sigma + O(\sqrt{\varepsilon})\Sigma_t , \qquad and \qquad \|\widehat{\Sigma} - \Sigma\|_F \leq O(\sqrt{\varepsilon})\|\Sigma_t\| .$$

We give the pseudocode for FIRSTPHASE in Algorithm 2. Our algorithm is simple: we simply run a naive pruning step on the tensored inputs, then apply Lemma 3.3 to the remaining tensored inputs.

---

**Algorithm 2** Robust Covariance Estimation With Bounded Covariance

1: **procedure** FIRSTPHASE($S = \{X_1, X_2, \ldots, X_n\}, \varepsilon, \Sigma_t$)&emsp;&emsp;&emsp;&emsp;&emsp;&emsp;▷ Theorem 4.6
2: &emsp;&emsp;**for** $i = 1, \ldots, N$ **do**
3: &emsp;&emsp;&emsp;&emsp;$Y_i \leftarrow \Sigma_t^{-1/2} X_i$
4: &emsp;&emsp;&emsp;&emsp;$Z_i \leftarrow Y_i \otimes Y_i$
5: &emsp;&emsp;**end for**
6: &emsp;&emsp;$S' \leftarrow$ NAIVEPRUNE($Z_1, \ldots, Z_N, 4d^2 N^2, 1/(dN \log \kappa)$)&emsp;&emsp;&emsp;&emsp;▷ Lemma 3.1
7: &emsp;&emsp;Let $\mathcal{O}_{\text{aug}} =$ APPROXIMATESCORE with $\delta = 1/\operatorname{poly}(d, \log \kappa)$
8: &emsp;&emsp;Let $\widetilde{\Sigma}$ be the estimate of $S'$ computed by Lemma 3.3 with score oracle $\mathcal{O}_{\text{aug}}$.
9: &emsp;&emsp;$\widehat{\Sigma} \leftarrow \Sigma_t^{1/2} \widetilde{\Sigma} \Sigma_t^{1/2}$.
10: &emsp;&emsp;$\Sigma_{t+1} \leftarrow \widehat{\Sigma} + O(\sqrt{\varepsilon})\Sigma_t$
11: &emsp;&emsp;**return** $\widehat{\Sigma}, \Sigma_{t+1}$
12: **end procedure**

---

*Proof of Theorem 4.6.* By Assumption 4.2 and Lemma 4.4, the points $Z_1, \ldots, Z_N$ are $O(\varepsilon, O(\sqrt{\varepsilon}), O(1))$-corrupted good with respect to $\Sigma_t^{-1/2} \Sigma \Sigma_t^{-1/2}$. Let $\sigma_t^2 = \|\Sigma_t^{-1/2} \Sigma \Sigma_t^{-1/2}\|$. Let $S'$ be the output of applying naive pruning to the $Z_i$. Observe that for all uncorrupted $i \in [N]$, we have that $\|Z_i\| \leq O(\sigma_t^2 d \log N)$. Thus, by the guarantee of Lemma 3.1, we have $\|Z_i\|_2 \leq O(\sigma_t^2 d \log N)$ for all $Z_i \in S'$, and $S'$ contains all remaining uncorrupted points in $[N]$.

Now, we have $S'$ satisfy all the conditions of Lemma 3.3 with $\sigma^2 = \sigma_t^2$ and $R \leq O(d\sigma^2 \log N)$. Let $\widetilde{\Sigma}$ be the estimation of $Z_i$ computed by Lemma 3.3 reshaped into $d \times d$ matrix. Condition on the event that the output of APPROXIMATESCORE is a valid output of an approximate augmented score

oracle in every iteration it is called in, which by our choice of parameters, occurs with probability $1 - 1/\operatorname{poly}(d, \log \kappa)$. In this event, by the guarantee on $\widetilde{\Sigma}$, we have

$$\|\widetilde{\Sigma} - \Sigma_Y\|_F = \|\widetilde{\Sigma} - \Sigma_t^{-1/2}\Sigma\Sigma_t^{-1/2}\|_F \le O(\sqrt{\varepsilon}).$$

This immediately implies $\widehat{\Sigma} - O(\sqrt{\varepsilon})\Sigma_t \preceq \Sigma \preceq \widehat{\Sigma} + O(\sqrt{\varepsilon})\Sigma_t$. Moreover, using the fact that $\|AB\|_F \le \|A\|\|B\|_F$, we have

$$\|\widehat{\Sigma} - \Sigma\|_F = \|\Sigma_t^{1/2}\widetilde{\Sigma}\Sigma_t^{-1/2} - \Sigma\|_F \le O(\sqrt{\varepsilon})\|\Sigma_t\|,$$

This proves the correctness guarantee. The runtime guarantee follows by combining Lemma 3.1, Lemma 3.3, and Theorem D.1. □

## D.3  Getting $O(\varepsilon \log 1/\varepsilon)$ error

We now turn our attention to SECONDPHASE, which allows us to refine a coarse estimate down to error $O(\varepsilon \log 1/\varepsilon)$. We restate the theorem below for convenience.

**Theorem 4.7** (Second Phase). *Let $S$ be a set of points satisfying Assumption 4.2. Let $0 < \zeta < \zeta_0$ for some universal constant $\zeta_0$. Given $\zeta_t$ and $\Sigma_t$ where $\Sigma \preceq \Sigma_t \preceq (1 + \zeta_t)\Sigma$ as input, Algorithm 3 runs in time $\widetilde{O}(T(N, d))$ and outputs a new upper bound matrix $\Sigma_{t+1}$ and a approximate covariance matrix $\widehat{\Sigma}$ such that, with probability $1 - 1/Nd$, for $\zeta_{t+1} = O(\sqrt{\varepsilon\zeta_t} + \varepsilon \log 1/\varepsilon)$, we have*

$$\Sigma \preceq \Sigma_{t+1} \preceq \Sigma + \zeta_{t+1}\Sigma_t , \qquad and \qquad \|\Sigma^{-1/2}\widehat{\Sigma}\Sigma^{-1/2} - I\|_F \le \zeta_{t+1} .$$

---

**Algorithm 3** Robust Covariance Estimation With Approximately Known Covariance

---

1: **procedure** SECONDPHASE($S = \{X_1, X_2, \dots, X_n\}, \varepsilon, \Sigma_t, \zeta_t$)      ▷ Theorem 4.7
2:     **for** $i = 1, \dots, N$ **do**
3:         $Y_i \leftarrow \Sigma_t^{-1/2}X_i$
4:         $Z_i \leftarrow Y_i \otimes Y_i$
5:     **end for**
6:     $S' \leftarrow$ NAIVEPRUNE($Z_1, \dots, Z_N, O(d \log N), 1/dN$)      ▷ Lemma 3.1
7:     Let $\mathcal{O}_{\mathrm{aug}} =$ APPROXIMATESCORE with $\delta = 1/\operatorname{poly}(d, N)$
8:     Let $\widetilde{\Sigma}$ be the estimation of $S'$ computed by Lemma 3.4.
9:     $\widehat{\Sigma} \leftarrow \Sigma_t^{1/2}\widetilde{\Sigma}\Sigma_t^{1/2}$.
10:     **return** $\widehat{\Sigma}$
11: **end procedure**

---

*Proof of Theorem 4.7.* The proof is very similar to that of Theorem 4.6. By the definition of $Y_i$ and condition on $\Sigma_t$, we know that

$$\|\Sigma_Y - I\| = \|\Sigma_t^{-1/2}\Sigma\Sigma_t^{-1/2} - I\| \le \zeta_t.$$

Since $\|\Sigma_Y - I\| \le \zeta$, by Lemma 4.4, the set $\{Z_1, \dots, Z_N\}$ is $O(\varepsilon, O(\varepsilon\sqrt{\log 1/\varepsilon}), O(\varepsilon\sqrt{\log 1/\varepsilon} + \zeta), O(\log 1/\varepsilon), O(\log^2 1/\varepsilon + \zeta)$-corrupted good with respect to $D_I$.

Therefore, $S'$ satisfies all condition of Lemma 3.4. Condition on the event that the output of APPROXIMATESCORE is a valid output of an approximate augmented score oracle in every iteration it is called. By Lemma 3.4 and our choice of parameters, this occurs with probability $1 - 1/\operatorname{poly}(d, N)$. Then, Lemma 3.4 guarantees that we can find a $\widetilde{\Sigma}$ where

$$\|\widetilde{\Sigma}^\flat - \mathbb{E}[Z_i]\|_2 = \|\widetilde{\Sigma} - \Sigma_Y\|_F \le O(\sqrt{\zeta_t\varepsilon} + \varepsilon \log 1/\varepsilon).$$

Then, we have

$$\|\Sigma^{-1/2}\widehat{\Sigma}\Sigma^{-1/2} - I\|_F = \|\Sigma^{-1/2}\Sigma_t^{1/2}\widetilde{\Sigma}\Sigma_t^{1/2}\Sigma^{-1/2} - I\|_F \le O(\sqrt{\zeta_t\varepsilon} + \varepsilon \log 1/\varepsilon),$$

where the last inequality follows by $\|AB\|_F \le \|A\|\|B\|_F$ and $\|\Sigma - \Sigma_t\| \le \zeta_t = O(1)$. This proves the correctness guarantee. Finally, the runtime guarantee follows by combining Lemma 3.1, Lemma 3.4, and Theorem D.1. □

# E  Fast Implementations for Tensor Inputs

In this section we describe how to implement the approximate score oracles described in Section 3.3.2 fast when the input is given as tensor products. Recall that an approximate score oracle takes as input a set of points $S$ of size $n$, and a sequence of weights $w_0, \ldots, w_{t-1}, w_t$, and computes $\tilde{\lambda}$ so that $\tilde{\lambda} \approx_{0.1} \|M(w_0) - I\|$, and $\tilde{\tau}_t \in \mathbb{R}^n$ where $\tilde{\tau}_{t,i} \approx_{0.1} \tau_{t,i}$ for all $i$ and $\tilde{q}_t$ where

$$\tau_{t,i} = (Z_i - \mu(w_t))^\top U_t (Z_i - \mu(w_t)) \ , \tag{6}$$

$$|\tilde{q}_t - q_t| \leq 0.1 q_t + 0.05 \|M(w_t) - I\|, \text{ where } q_t = \langle M(w_t) - I, U_t \rangle \ , \tag{7}$$

and

$$U_t = \exp\left( c I_d + \frac{1}{1.1\alpha} \sum_{i=0}^{t-1} M(w_i) \right) = \frac{\exp\left( \alpha \sum_{i=0}^{t-1} M(w_i) \right)}{\operatorname{tr} \exp\left( \alpha \sum_{i=0}^{t-1} M(w_i) \right)} \ ,$$

where $\alpha > 0$ is a parameter, and $c$ is chosen so that $\operatorname{tr}(U_t) = 1$.

## E.1  Several Ingredients

To implement the approximate score oracle efficient, we need several ingredients. The first one is Taylor series approximation for the matrix exponential:

**Lemma E.1** ([AK16]). *Let $0 < \varepsilon < 1$, let $X$ be a PSD matrix where $\|A\| < M$, there is a degree-$\ell$ polynomial $P_\ell$, where $\ell = O(\max(M, \log 1/\varepsilon))$, such that*

$$(1 - \varepsilon) \exp(X) \preceq P_\ell(x) \preceq (1 + \varepsilon) \exp(X).$$

Another difficulty is that writing down the matrix $U_t$ takes the time $\Omega(d^4)$. Then, we need the Johnson-Lindenstrauss Lemma [JL84] to construct a matrix in much lower dimension:

**Lemma E.2** (Johnson-Lindenstrauss Lemma (JL Lemma)). *Let $J \in \mathbb{R}^{r \times d}$ be a matrix whose each entries are i.i.d. samples from $\mathcal{N}(0, 1/r)$. For every vector $v \in \mathbb{R}^d$ and every $\varepsilon \in (0, 1)$,*

$$\Pr[\|Jv\|_2 \approx_\varepsilon \|v\|_2] > 1 - \exp(-\Omega(\varepsilon^2 r)).$$

**Lemma E.3** (Tellegen's Theorem, [BCS97]). *Fix a matrix $A \in \mathbb{R}^{r \times c}$, if we can compute matrix-vector product $Ax$ for any $x \in \mathbb{R}^c$ in time $t$. Then, we can compute $A^\top y$ for any $y \in \mathbb{R}^r$ in time $O(t)$.*

**Lemma E.4** (Power method). *For any matrix $A \in \mathbb{R}^{m \times m}$, there exists an randomized algorithm, with probability $1 - \delta$, outputs its $1 \pm \varepsilon$ approximation using $O(\log m \log(1/\delta)/\varepsilon)$ many matrix-vector multiplications.*

## E.2  Fast approximate score oracle

Observe that $\mathscr{O}_{\text{aug}}$ is strictly more powerful than $\mathscr{O}_{\text{approx}}$. In this section, we describe how to implement $\mathscr{O}_{\text{aug}}$ fast.

**Lemma E.5.** *Assuming $\alpha$ is chosen such that it always satisfies Equation* (3)*, then* APPROXIMATESCORE$(S, w_1, \ldots, w_t)$ *runs in time $\widetilde{O}(t^2 \cdot T(N, d) \log 1/\delta)$.*

First, we show how to compute $M(w_i)v$ for any $v$ by utilizing the fast rectangular matrix multiplication.

**Lemma E.6.** *For any vector $v \in \mathbb{R}^{d^2}$ and $w_i \in \mathbb{R}^N$. We can compute the matrix-vector product $M(w_i) \cdot v$ in $O(T(N, d))$ time.*

*Proof.* Let $Z \in \mathbb{R}^{d^2 \times N}$ be the matrix whose $i$-th column is $Z_i$ and $Y \in \mathbb{R}^{d \times N}$ be the matrix whose $i$-th column is $Y_i$. Note that $M(w_i) = (Z - \mu(w_i)\mathbf{1}^\top) \operatorname{diag}(w_i/|w_i|)(Z - \mu(w_i)\mathbf{1}^\top)^\top$.

By Lemma E.3, $(Z - \mu(w_i)\mathbf{1}^\top)^\top v$ has the same running time as $(Z - \mu(w_i)\mathbf{1}^\top)v$. We have $(Z - \mu(w_i)\mathbf{1}^\top)v = Zv - (\mathbf{1}^\top v)\mu(w_i)$. We observe that $Zv = (Y \operatorname{diag}(v)Y^\top)^\flat$, then we can compute $Zv$ in $O(T(N, d))$ time. Then we can compute $\operatorname{diag}(w_i/|w_i|)(Z - \mu(w_i)\mathbf{1}^\top)^\top v$ in time $O(T(N, d))$ since multiply a diagonal matrix by a vector can be done in $O(N)$. Thus, we can compute $M(w_i)v$ for any $v$ in time $O(T(N, d))$. $\qquad\square$

---

**Algorithm 4** Approximate Score Oracle

---

1: **procedure** APPROXIMATESCORE($\delta, \alpha, S = \{Y_1, \ldots, Y_N\}, w_1, w_2, \ldots, w_t$)
2:     $r \leftarrow O(\log 1/\delta), \ell \leftarrow t$
3:     Let $J \in \mathbb{R}^{r \times d^2}$ matrix whose each entries are i.i.d. samples from $\mathcal{N}(0, 1/r)$.
4:     Let $Z_i = Y_i \otimes Y_i$ and $\mu(w_i) = \frac{1}{|w_i|} \sum_{j=1}^{N} w_{i,j} Z_j$,
5:     Let $M(w_i) = \frac{1}{|w_i|} \sum_{j=1}^{N} w_{i,j} (Z_j - \mu(w_i))(Z_j - \mu(w_i))$ and $A = \frac{\alpha}{2} \sum_{i=0}^{t-1} M(w_i)$.
6:     Compute $S_{r,\ell}$ where $S_{r,\ell} = J \cdot P_\ell(A)$
7:     Compute $\nu = \text{tr}(S_{r,\ell} S_{r,\ell}^\top)$.
8:     Compute $B = S_{r,\ell}(Z - \mu(w_t)\mathbf{1}^\top)$
9:     **for** $i = 1, \ldots, N$ **do**
10:         Let $\tilde{\tau}_{t,i} = \frac{1}{\nu} \|B_{:i}\|_2^2$
11:     **end for**
12:     Let $\tilde{q}_t = \sum_{i=1}^{N} (\tilde{\tau}_{t,i} - 1)$.
13:     Compute $\tilde{\lambda} \approx_{0.1} \|M(w_t) - I\|$ using power method with $r$ many iterations.
14:     **Output** $\tilde{\tau}_t, \tilde{q}_t, \tilde{\lambda}$.
15: **end procedure**

---

*Proof of Lemma E.5.* By Lemma E.6, we can compute $Av$ for any $v$ in time $O(tT(N,d))$. By repeatedly multiplying $A$ on the left, we can compute $A^k v$ in time $O(k \cdot tT(N,d))$. Since we take $\ell = O(t)$, we can compute $P_\ell(A)v$ for any $v$ in time $\widetilde{O}(t^2 \cdot T(N,d))$. We can compute $J \cdot P_\ell(A) = (P_\ell(A)J^\top)^\top$ by multiply each column of $J^\top$ to $P_\ell(A)$. Thus, $S_{r,\ell}$ can be computed in time $\widetilde{O}(r \cdot t^2 \cdot T(N,d)) = \widetilde{O}(t^2 \cdot T(N,d))$. We can compute $Z^\top v$ in $O(T(N,d))$ by multiplying each row of $S_{r,\ell}$ with $Z$. Therefore, we can compute matrix the $B$ in $\widetilde{O}(T(N,d))$.

Now, we consider how to compute $\nu$. Note that $\nu = \text{tr}(S_{r,\ell} S_{r,\ell}^\top) = \sum_{i=1}^{r} (S_{r,\ell} S_{r,\ell}^\top)_{i,i} = \sum_{i=1}^{r} \|S_{r,\ell} e_i\|_2^2$. Thus, $\nu$ can also be computed in time $\widetilde{O}(T(N,d))$. Once we have $B$ and $\nu$, then we can compute $\tilde{\tau}$ and $\tilde{q}$ in $\widetilde{O}(N)$ time.

Using power method (Lemma E.4), we can find a $1 \pm 0.1$ approximation of $\|M(w_t) - I\|$ using $O(\log d \log 1/\delta)$ matrix-vector multiplications. By Lemma E.6, we can compute $(M(w) - I) \cdot v$ for any $v$ in time $O(T(N,d))$. Then, the total runtime is $O(T(N,d)) \cdot O(\log d \log(1/\delta)) = \widetilde{O}(T(N,d) \log(1/\delta))$.

Thus, the algorithm runs in time $\widetilde{O}(t^2 \cdot T(N,d) \log 1/\delta)$. $\qquad\qquad\square$

**Lemma E.7.** *The output of* APPROXIMATESCORE$(S, w_1, \ldots, w_t, \alpha)$ *satisfies* $\tilde{\tau} \approx_{0.1} \tau$ *and* $\tilde{q} \approx_{0.1} q$ *with probability* $1 - \delta$.

The correctness proof directly follows by the original correctness proof in [DHL19]; for completeness, we prove it below.

*Proof.* We condition on two events occurring. Let $\tilde{\lambda}$ be the output of Line 13 in Algorithm 4.

- $\|J \cdot P_\ell(A)v\|_2 \approx_{0.01} \|P_\ell(A)v\|_2$ for all $v \in \{e_1, \ldots, e_{d^2}\} \cup \{X_1 - \mu(w_t), \ldots, X_N - \mu(w_t)\}$.

- $\tilde{\lambda} \approx_{0.1} \lambda$.

Note that by our choice of parameters, both events occur individually with probability at $1 - \frac{\delta}{3}$. Then, by a union bound over failure probability, these two events succeeds with probability at $1 - \delta$.

The guarantee on $\tilde{\lambda}$ directly follows from correctness of power method. Now, we show $\tilde{\tau} \approx_{0.1} \tau$. Let $M = \frac{\alpha}{2} \sum_{i=1}^{t-1} M(w_i)$. Then, we have

$$\tau_i = (X_i - \mu(w_t))^\top \frac{\exp(2M)}{\text{tr}\exp(2M)}(X_i - \mu(w_t)) = \frac{1}{\text{tr}\exp(2M)} \|\exp(M)(X_i - \mu(w_t))\|_2^2,$$

where as $\tilde{\tau}_i = \frac{1}{\mathrm{tr}(S_{r,\ell}S_{r,\ell}^\top)}\|S_{r,\ell}(X_i - \mu(w_t))\|_2^2$.

Note that

$$\|S_{r,\ell}(X_i - \mu(w_t))\|_2^2 = \|J \cdot P_\ell(A)(X_i - \mu(w_t))\|_2^2 \approx_{0.01} \|P_\ell(A)(X_i - \mu(w_t))\|_2^2$$
$$\approx_{0.03} \|\exp(M)(X_i - \mu(w_t))\|_2^2, \quad (8)$$

where the first line follows by Lemma E.2 and the last line follows by our choice of $\ell$ and Lemma E.1.

Similarly, we have $\exp(2M)_{i,i} = \|\exp(M)e_i\|_2^2$ and $(S_{r,\ell}S_{r,\ell}^\top)_{i,i} = \|S_{r,\ell}e_i\|_2^2$.

By definition of $S_{r,\ell}$, we have

$$\|S_{r,\ell}e_i\|_2^2 = \|J \cdot P_\ell(A) \cdot e_i\|_2^2 \approx_{0.01} \|P_\ell(A) \cdot e_i\|_2^2 \approx_{0.03} \|\exp(M)e_i\|_2^2$$

and this immediately implies $\mathrm{tr}(S_{r,\ell}S_{r,\ell}^\top) \approx_{0.03} \mathrm{tr}(\exp 2M)$. Thus, we have $\tilde{\tau}_i \approx_{0.07} \tau_i$.

Now, we show $\tilde{q}$ is close to $q$. Rewriting $\tilde{q}$, we get

$$\tilde{q} = \frac{1}{\mathrm{tr}(S_{r,\ell}S_{r,\ell}^\top)} \sum_{i=1}^N (\|S_{r,\ell}(X_i - \mu(w_t))\|_2^2 - \mathrm{tr}(S_{r,\ell}S_{r,\ell}^\top))$$

$$= \frac{1}{\mathrm{tr}(S_{r,\ell}S_{r,\ell}^\top)} \left\langle M(w_t) - I, P_\ell(M)JJ^\top P_\ell(M) \right\rangle$$

$$= \frac{1}{\mathrm{tr}(S_{r,\ell}S_{r,\ell}^\top)} (\langle M(w_t) - I, \exp(2M) \rangle + \xi),$$

where $|\xi| \leq 0.02\|M(w_t) - I\| \cdot \mathrm{tr}\exp(2M)$. We complete the proof by note that $\mathrm{tr}(S_{r,\ell}S_{r,\ell}^\top) \approx_{0.03} \mathrm{tr}(\exp 2M)$. $\qquad\square$