[Reviews · NeurIPS 2020]

Review 1

Summary and Contributions: The paper studies the problem of robust covariance estimation, i.e, given N samples from N(0,\Sigma), of which \epsilon fraction are contaminated, the goal is to design an estimator \hat{\Sigma} which recovers the true covariance \Sigma in the Mahalanobis norm. For this problem, the authors provide the first polynomial time algorithm which gets the (near-)optimal statistical error, and runs in near-matrix multiplication time, i.e. O(T(n,D) \log(\kappa)), where \kappa is the condition number and T(n,D) is time taken to multiply d \times N with its transpose. Note that previous methods obtaining similar statistical guarantees have an additional poly(1/\epsilon) dependence, and removing that is the main contribution of the paper.

Strengths: The paper is a strong theoretical work. Technically, the paper hinges on combining ideas from the quantum entropy scoring algorithm of [DHL19] with the existing covariance estimation algorithm in [CDGW19]. The problem itself is very relevant to the community and I believe that the idea should be useful in other problems as well.

Weaknesses: Dependence on condition number. Is there any intuition on why this condition number dependence appears, and how can one possibly remove it, if possible? It would be good to do actual runtime comparisons and experiments between CDWG19 and this work to demonstrate the improvement in run time. ### After Rebuttal #### I thank the authors for their thoughtful response. After looking at it and reading the other reviews, my score remains the same.

Correctness: Yes.

Clarity: Yes.

Relation to Prior Work: Yes.

Reproducibility: Yes

Additional Feedback: Minor Point: Note that the main submission file also includes the supplementary material, and in particular, violates the submission policy.


Review 2

Summary and Contributions: This paper extends recent lines of work for achieving nearly linear time algorithms for robustly estimating core statistical quantities (e.g., mean and covariance of Gaussians). The main result is a nearly linear time algorithm for robust covariance estimation of a gaussian that has no dependence on the error parameter \epsilon.

Strengths: This is a very nice addition to the literature. The problem is fundamental and the end result is impressive. The paper seems correct and well written.

Weaknesses: The paper seems to import most of its machinery from a prior work [DHL] on quantum entropy scoring. In particular, it seems as though some very technical details need to be adjusted to suitably modify work due to CDGW19.

Correctness: Yes

Clarity: Yes

Relation to Prior Work: Yes

Reproducibility: Yes

Additional Feedback: I think the paper should be accepted, but it seems less exciting than a similar prior work of DHL that pioneered most of the novelty (applied in that case only to robust mean estimation).


Review 3

Summary and Contributions: The main contribution of the paper is to introduce an algorithm that improves time complexity while maintains the same error as reference [CDGW19]. The authors achieve this by replacing packing SDP phase in [CDGW19] With an approximated quantum entropy score filtering.

Strengths: This is a relevant work to the community and of significant interest. The work also has theoretical grounding and shows that the time complexity of the corrupted samples is almost the same as clean samples, thus it is robust to corruption with the same time complexity.

Weaknesses: The reviewer thinks it would be more interesting if the authors could also provide simulations to support their results.

Correctness: I have not gone through the details of the proof but the general claims and method sounds correct to me.

Clarity: The paper is generally written acceptably but can be improved.

Relation to Prior Work: The reviewer thinks it is not explained enough that how the authors are getting the better time complexity.

Reproducibility: Yes

Additional Feedback: In line 38, it will be better to provide a reference for the information theoretic bound on error for covariance estimation. In line 39, “However, …”, please provide a reference or clarify why the previous algorithms achieve O(\epsilon) error that its run time complexity is exponential in 'd'. The reviewer thinks although the total variation distance is well known but $d_{TV}$ in line 110 needs clear definition. Also the reviewer thinks it is nice if 'd' be defined after the abstract in the introduction before the first appearance as well. Please provide an intuition for the regularity condition in Definition 3.2. There is a typo in the probability of Lemma 4.3. In Assumption 4.2, please clarify what are S_b and S_r.


Review 4

Summary and Contributions: The paper considers the problem of robust covariance estimation. In the usual covariance estimation problem, given samples from d-dimensional Gaussian N(0, Sigma), the goal is to use as few samples as possible to recover the covariance matrix Sigma. In the robust version, an eps-fraction of the samples is arbitrarily corrupted and the goal is still to recover the covariance matrix Sigma. The error of an estimate Sigma’ is given by |Sigma^{-1/2} * Sigma’ * Sigma^{-1/2} - I|_F. The best possible error is O(eps) with O(d^2/eps^2) samples---which is tight---but the algorithm is not polynomial time. The previous best polynomial-time algorithm achieves O(eps log(1/eps)) with O(d^2/eps^2), with a bad eps-dependence of 1/eps^8 in the runtime. This paper removes the dependence on eps in the runtime while maintaining the O(eps log(1/eps)) error.

Strengths: Removal of 1/eps^8 in runtime while maintaining the best error achieved by polynomial-time algorithms.

Weaknesses: Techniques were developed in earlier works. This paper plugs the new technique (based on the QUE score) for mean estimation in [DHL19] into the previous result on covariance estimation [CDGW19]. [Previously I asked why it only worked for Gaussians with mean zero because I thought it should work for an arbitrary Gaussian. The author has confirmed this in their rebuttal.]

Correctness: The results seem correct to me though I didn't check all the details.

Clarity: The paper is largely well-written. I think the authors can do more highlighting on the difference from [CDGW19]. It seems that the sketch of Algorithm 1 is completely the same as in [CDGW19], even the guarantee of each step is the same. This should be made clearer. [I agree with the authors' rebuttal that the difference lies in the fine-tuning. I still think, in respect of writing, the authors are not doing enough to mention how much this paper inherits from [CDGW19]. ] Discussion on the running time of the approximate score oracle (Theorem D.1) should also be included in the main body. This is important and should not be swept under the carpet. In Line 11 of Algorithm 1, shouldn’t it \hat\Sigma_{T_2+1}?

Relation to Prior Work: The difference is the result is clearly discussed but the difference in approach was not thoroughly discussed. I think more highlight on the difference is needed.

Reproducibility: Yes

Additional Feedback:

[Author Response · NeurIPS 2020]

We thank the reviewers for their thoughtful reviews. Before we begin, we apologize for accidentally including the
supplementary material in the main text. This was an honest mistake at submission time. We do not believe it changes
anything in the review process, since the exact same supplementary material is included anyways, and we hope that the
reviewers and ACs can overlook it.

**General remarks** We begin by addressing some shared questions from the reviewers.

Some of the reviewers were interested in seeing simulation studies for our algorithm. We agree with the reviewers
that experimental evaluations would be interesting, and would be very useful follow-up. In our work, we focus on
developing the theoretical foundations, which we believe are already valuable to the community. However, we do
believe our algorithm is eminently practical, as it does not require any heavy-duty theoretical tools. In contrast, the
algorithm in [CDGW19] is quite complex, and uses some complex optimization techniques which are likely impractical.

Some of the reviewers also had questions about the relationship of our work with previous ones, specifically [CDGW19]
and [DHL19]. We will clarify the relationship more thoroughly in the next version of the paper, and we will also briefly
explain here. At a very high level, we follow the same algorithm structure as [CDGW19], e.g. in Alg 1, and we use
their subroutine to get a crude estimate of the covariance, e.g. in Alg 2. However, most of the technical work in both
papers lies within the fine-tuning step (Alg 3 in our paper), and here we use very different tools. [CDGW19] relies on
very heavy-duty optimization primitives, specifically, nearly-linear time packing SDP solvers. In contrast, we design
much more streamlined and specialized tools to solve our optimization problems, which exploit the structure of the
problem much more effectively. Morally, this is why we are able to improve the runtime of [CDGW19].

We do directly build off of the framework in [DHL19]. However, there are a number of non-trivial technical challenges
to adapt their framework—which was developed for mean estimation—to the covariance estimation setup, which we
must overcome. We also believe that it is an interesting contribution in its own right to demonstrate the generality of the
QUE scoring framework. In particular, we found it somewhat surprising that it is able to produce arguably one of the
cleanest algorithms for such a fundamental problem in robust statistics, which is simultaneously optimal in terms of
runtime, sample complexity, and error (up to logarithmic factors).

**Reviewer specific comments** We thank all the reviewers for detailed comments, and we will fix the minor issues the
reviewers pointed out in the next version. We now address individual reviewer's questions and comments that were not
addressed above.

Reviewer 1: The $\log \kappa$ comes from our invocation of of Lemma 3.1 in [CDGW19], which is used to get a rough estimate
the covariance. Intuitively, this $\log \kappa$ comes from the fact that first order methods such as matrix multiplicative weights
typically must pay a logarithmic factor in the "width" of the problem, and the only width bound we can typically obtain
a priori depends on $\kappa$. It is an interesting theoretical question to avoid paying this factor with linear-time algorithms
(note that slower polynomial time algorithms do not have to pay this factor). We believe this is a mild cost, since we
only depend logarithmically on $\kappa$, and in almost all applications, the condition number is polynomially bounded.

Reviewer 2: We thank the reviewer for their kind words. Please see discussion above related to their comments regarding
simulation and comparison to prior work.

Reviewer 3:
• As briefly mentioned above, the speed-up comes from designing better optimization routines that heavily utilize the
structure of the problem. At a very high level, while the width of the problem does scale with $\varepsilon$, we demonstrate
that the uncorrupted data satisfies the regularity condition (Def. 3.2) actually allows us to pretend that the width
is constant, even when we are running matrix multiplicative weights to high precision. We will explain this more
thoroughly in the next version of the paper.
• Def 3.2 says that not only do the empirical mean and covariance of the data concentrate, but any large subset of the
data still has good concentration. Intuitively, we should expect this to be true because Gaussians exhibit very strong
concentration, and this still holds when we condition on events with probability $1 - \varepsilon$.

Reviewer 4:
• As discussed in lines 127-130, our results hold (at a cost of doubling $\varepsilon$) for Gaussians with arbitrary mean. As
mentioned in lines 58-59, by combining this with previous work, this allows us to learn arbitrary Gaussians to good
total variation distance.
• Please see previous discussion in regards to the relationship to [CDGW19].
• We agree that the approximate score oracles are important and we discuss them at a high level in Sec. 3.3.2. However,
the details are quite technical, and due to space considerations we chose to move the formal discussion to the
supplementary material, in favor of presenting (what we consider to be) the main takeaways in the main text.

[Meta-Review · NeurIPS 2020]

This paper gives a nearly linear time algorithm for estimating the covariance of an unknown Gaussian distribution in the presence of a small constant \epsilon fraction of outliers. This has been an important research direction in the past few years and recent works used the idea of "quantum entropy scoring" to design a similar linear time algorithm for estimating the mean. The main contribution of this paper is extending that idea to the harder task of covariance estimation. One of the concerns in the discussion was the relationship to the prior works on robust linear time algorithms for mean estimation. The reviewers found the authors' feedback on this useful and clarifying. I recommend accepting this paper to the NeurIPS program.